# Systematic investigation of inadequate food access at a large southeastern land grant university

**Ralph P. Hall** [1]*, **Jessica Agnew**[2], **Wei Liu**[3], **Lana Petrie**[4], **Chris North**[3]

**1** The School of Public and International Affairs, Virginia Tech, Blacksburg, Virginia, United States of America, **2** CALS Global, Virginia Tech, Blacksburg, Virginia, United States of America, **3** Department of Computer Science, Virginia Tech, Blacksburg, Virginia, United States of America, **4** College of Agriculture, Tennessee State University, Nashville, Tennessee, United States of America

* rphall@vt.edu

**Data Availability Statement:** All relevant data are available from the OSF repository (https://doi.org/10.17605/OSF.IO/BNUQV).

**Funding:** The initial research that made this study possible was supported by funding from the

## Abstract

Over the past decade, the impact of low food security on student well-being and academic performance has become a growing concern at institutions of higher education across the U.S. This mixed methods study adds to the growing body of evidence on the association between student socio-demographic and economic characteristics and food security. An online survey covering food access, student well-being, and housing security was sent to 35,337 undergraduate and graduate students at a large southeastern land grant university. A total of 2,116 complete responses were received; a 6% response rate. The survey responses also included 176 written statements by students. The survey found that 16% of both undergraduate and graduate students had low or very low food security, as defined by a modified USDA food security measure. The socio-demographic and economic characteristics that were linked to a higher likelihood of low food security included: having a GPA of less than 3.0, having a disability, being an international student, being a first-generation student, being a transfer student, going into debt to pay for food, being a Black or African American student, having poor mental health, having uncertain living arrangements, and having no medical insurance. Recommendations for enhancing student access to food, housing, and mental health services are discussed.

## Introduction

Food accessibility at institutions of higher education has become a growing concern over the past decade in the United States [1–4] and in other countries [5–7]. Since the opportunity to attend college can be viewed as a privilege, concerns such as food access, housing security, and well-being are often considered secondary issues or overlooked altogether [8]. However, up to 50% of student populations, depending on the geographic location and institution, can be classified as having very low or low food security [9]. A growing number of studies are also focusing on the broader issue of food access (which encompasses food security), housing security, and well-being [5,10–12]. Mounting evidence on the impact these experiences have on student

Virginia Tech Center for International Research, Education, and Development (CIRED), the office of Outreach and International Affairs (OIA), and the College of Science. These entities had no role in the study design, data collection and analysis, decision to publish, or preparation of the manuscript. The funders had no role in study design, data collection and analysis, decision to publish, or preparation of the manuscript.

**Competing interests:** The authors have declared that no competing interests exist.

performance [8,10,11,13,14] and quality of life [5,15] have brought these concerns to the attention of higher education administrations across the nation.

To ensure that institutions of higher education offer their students a food secure experience, it is critical that research identifies students who may be at risk of food insecurity [16–18]. This can be difficult to achieve, as a lack of access to safe, affordable, nutritious, and culturally relevant foods can be transient in the college environment, is sometimes considered a 'rite of passage' by students and their parents, and is multi-dimensional and complex [19–21]. There are also many underlying factors or characteristics that may put a student at risk of food insecurity that are often unobservable to university staff and faculty [22,23]. Furthermore, methodological standards and indicator calculation may obfuscate the students most at risk or currently experiencing food insecurity. For example, when measuring poverty in the United States, college students are excluded from the estimates if they live in a residence hall because the population is transient [19,24]. There is also mixed evidence on the validity of the 10-item USDA Household Food Security instrument (with screener question) to assess food security among college students in the U.S. context [25–27]. However, it has been found to perform better than the 6-item version with regards to measuring food security among the college student population [28].

Recent surveys on the prevalence of food security at four-year institutions of higher education using the USDA instrument estimate that between 24% to 40% of students have low or very low access to food [29]. The rates of food insecurity at two-year institutions tend to be higher and fall between 35% to 49% [29]. Other systematic reviews of food insecurity studies find average rates of food insecurity of 35% [9] and 41% [30]. However, the heterogeneity of the studies' methods and differing student populations and geographic locations make it difficult to draw comparisons.

Recent research explores a broad range of negative impacts associated with inadequate food access on college campuses [8,10,13,31]. For example, the association between food insecurity and poor mental health outcomes such as increased depression, anxiety, and stress, which can also impact sleep, has been well documented [15,32–34]. Leung et al. [11] found that if a student experiences food, financial, or housing insecurity they were more likely to have anxiety, depression, fair/poor health, and a lower mean GPA than students who were secure in these dimensions. In one of the first longitudinal studies on food insecurity and college graduation rates, Wolfson et al. [31] found that students who experienced food insecurity during college were associated with lower odds of completing their degree. Further, if these students did complete their education, they were more likely to obtain an associate's rather than a bachelor's degree, which could have a long-lasting impact on their future economic mobility.

Demographic and economic characteristics are associated with different levels of food security as well. For example, studies have found that if a student is not white, is a first-generation student, experiences housing insecurity, and/or receives financial aid, they were more likely to experience food insecurity while in college [10,31,35–37]. Students accumulating loans/debt are also more at risk of being food insecure [38]. A qualitative study by Martinez et al. [12] enriches these findings by exploring the interconnected nature of food and housing insecurity and introduces other factors such as transportation barriers that can limit a student's access to food. They also expand the group of nontraditional students at risk of being food insecure to include out-of-state, international, mature students, and students with dependents. In many cases, students in need were found to have difficulty navigating financial aid, the aid that was provided is insufficient, or they simply do not qualify for financial assistance, revealing the gaps needing to be filled.

The COVID-19 pandemic also worsened the situation at many institutions of higher education [39,40]. Soldavini et al. [41] examined how COVID-19 impacted the food security status

of students and what characteristics were associated with changes to food security status. Their study showed how food insecurity increased for students during the pandemic due to job loss or the need to provide financial support to their family. Similar results were found by Hagedorn et al. [42]. Soldavini et al. [41] identified a positive impact on food security status if a student had moved back home with their family or had received financial support from family members before and/or during the pandemic. Owens et al. [43] found that students who had a change in employment status or living arrangements during the pandemic were more likely to be food insecure and students living with parents or relatives were less likely to be food insecure compared to students living alone.

While the evidence of food security on college campuses is growing and a few recent studies have better characterized the portion of student populations that experience food insecurity, there is still potential to expand our understanding of associated factors and experiences that prevent students from accessing the food they need physically, culturally, and socially. Such knowledge will help to more effectively identify students at risk and develop the systems to ensure that students' needs are adequately met while pursuing an education. Mixed methods and standardized approaches are needed to assist university administrations in identifying these students as well as the structural factors that create barriers to accessing the food and resources needed to have a successful university experience.

Following the lead of Martinez et al. [12], this systematic investigation used mixed methods to further illuminate the range of factors that contributes to food insecurity among the student body at a large southeastern land grant university. The objectives were to: (i) compare the students' socio-demographic and economic characteristics associated with food insecurity versus food security; (ii) identify correlates related to wellbeing, housing, and COVID-19; and (iii) understand students' interpretation of food insecurity and barriers to accessing safe, nutritious, affordable, and culturally relevant foods.

## Methods

In 2021, an online survey focused on food security, student well-being and mental health, and housing was sent to all 35,339 undergraduate and graduate students enrolled at a large southeastern land grant university. The survey responses were collected confidentially using a Qualtrics link distributed via email. A total of four recruitment emails were sent between April 5 to 30, 2021. The students were not offered any incentives to participate. To be included in the study, students needed to be over the age of 18 and enrolled either as a part- or full-time student. Students who did not meet these criteria were excluded. After reading the informed consent script at the beginning of the survey, students were asked whether they agreed to participate in the study. Those who selected "yes" were able to proceed, and those who selected "no" were sent to the end of survey and thanked for their time.

The survey used the USDA's 10-item Household Food Security Survey Module (FSSM) with a screener question (Table 1) [44]. Since its creation in 1995, this instrument has been subject to rigorous [45,46] and ongoing technical review, which has led to minor adjustments being made to the questions [47]. As a result, the instrument is considered to provide a valid and reliable food security estimate of a population and is included in a range of U.S. national surveys such as the Current Population Survey (CPS) [47]. In 2019, Nikolaus et al. [28] found that the 10-item USDA FSSM with screener questions performed the best in terms of model fit among the college student population when compared with the 6-item FSSM without screener questions. Of the available USDA tools to measure food security, the 10-item FSSM was the leading instrument at the time the survey was conducted. Further, the 12-month recall version of the 10-item FSSM was used since the 30-day recall instrument is intended for shorter-term program evaluation.

**Table 1. Food security survey module.**

| Question | Response (points awarded) |
|---|---|
| **Screener**: Which of these statements best describes the food eaten by you (or your dependents) in the last 12 months | I have enough to eat and the kinds of food I want [**Exit module**] <br> I have enough to eat but not always the kinds of food I want / Sometimes I don't have enough to eat / Often, I don't have enough to eat [**Continue**] |
| For you or your household (your dependents, including spouse, partner, and/or children), In the last 12 months . . . | |
| . . . I worried whether my food would run out before I got money to buy more | Often True / Sometimes True (**1**) <br> Never True / Don't Know / Prefer not to say (**0**) |
| . . . The food I bought didn't last and I didn't have money to buy more | Often True / Sometimes True (**1**) <br> Never True / Don't Know / Prefer not to say (**0**) |
| . . . I couldn't afford to eat balanced meals (i.e. contained each of the 5 food groups—vegetables, fruits, dairy, protein, and grains) | Often True / Sometimes True (**1**) <br> Never True / Don't Know & Prefer not to say (**0**) |
| . . . Did you ever eat less than you felt you should because there wasn't enough money for food? | Yes, almost every month / Yes, some months but not every month / Yes, only 1 or 2 months (**1**) <br> No / Don't know (**0**) |
| . . . Were you ever hungry but didn't eat because there wasn't enough money for food? | Yes, almost every month / Yes, some months but not every month / Yes, only 1 or 2 months (**1**) <br> No / Don't know (**0**) |
| . . . Did you ever cut the size of your meals, or skip meals because there wasn't enough money for food? | Yes, almost every month / Yes, some months but not every month (**2**) <br> Yes, only 1 or 2 months (**1**) <br> No / Don't know (**0**) |
| . . . Did you or any other adults in your household (not including roommates) ever not eat for a whole day because there wasn't enough money for food? | Yes, almost every month / Yes, some months but not every month (**2**) <br> Yes, only 1 or 2 months (**1**) <br> No / Don't know (**0**) |

The word household was removed from the screener question and respondents were asked instead about their experiences and, if applicable, about their dependents. This change was made to reduce any confusion around what constituted a 'household' for students living with roommates. While text referencing "your household" was retained in the recall questions, it was qualified to mean "dependents, including spouse, partner, and/or children." Further, respondents were asked to exclude roommates for the question asking about "other adults in your household." Table 1 displays the questions asked across the three stages of the instrument, and indicates the points that were awarded for a certain response to each question. The USDA question asking respondents "In the last 12 months, did you lose weight because there wasn't enough money for food?" was removed from the module. This question is problematic since students may gain weight since they do not have access to nutritious foods [14]. Further, those studies that have attempted to understand the relationship between food security and weight have found mixed results. For example, El Zein et al. [33] found no significant difference between students with low versus high food security and their body mass index (BMI) and waist circumference. Whereas, Martinez et al. [48] found direct and indirect links between low food security and increased BMI and poor health.

Food security status was ascertained by scoring the responses as outlined in Table 1 and categorizing the respondent into one of four segments (Table 2). The analysis dichotomizes respondents into 'high' (consisting of students with high or marginal) and 'low' (consisting of students with low or very low) food security status, a common approach among food security studies. The removal of the weight loss question means the USDA metric used in this study has a range of 1 to 9 (rather than 1 to 10).

**Table 2. Food security categories.**

| Food Security Score | Food Security Status | Interpretation |
|---|---|---|
| 0 | High Food Security | Respondent has no reported indications of barriers or limitations of food-access. |
| 1–2 | Marginal Food Security | Respondent has one or two indications of food insecurity typically in terms of anxieties over insufficient food in the house. There are little to no changes in diets or food access. |
| 3–5 | Low Food Security | Respondent has some indications of food insecurity–reports of reduced diet quality, variety, and/or desirability. Little to no indication of reduced food intake. |
| 6–9 | Very Low Food Security | Respondent has multiple indications of food insecurity including disrupted eating patterns and reduced food intake. |

Version 3.6.3 of the R programming language was used for statistical analysis. Odds ratios (OR) for selected socio-demographic or economic characteristics of interest were calculated. When the OR is greater than 1, it indicates the increased occurrence of low food security; when OR is less than 1, the occurrence of low food security decreases [49,50]. Odds ratios of = 1 or confidence intervals that include 1 are insignificant. Odds ratios were estimated using a Generalized Linear Model (GLM).

A Chi-square analysis was used to assess if there was a statistically significant difference between the proportions of students who were in the low versus high food security status. This test was used as the variables were categorical. A permutation chi squared test was conducted when the sample size in any individual group was smaller than or equal to 5. Statistical significance was considered at $p < 0.05$.

Open-ended answers were analyzed following Thomas and Harden's [51] thematic synthesis approach. Three researchers simultaneously conducted an inductive thematic coding of the qualitative responses and then engaged in a group discussion of overlaps and divergences. Given the importance and sensitivity of the themes, this approach more accurately categorizes the issues that contribute to food insecurity, when compared with quantifying the degree of consensus by using an intercoder consistency or intercoder reliability rating. Coding was conducted in Word using colored highlighting and textual coding.

The study was undertaken in compliance with the institutional research protocol IRB-21-122.

## Respondent characteristics

A total of 2,116 students completed the entire survey for a respondent rate of 6%. The majority of the 2,116 respondents (94%) were enrolled at the university's main campus in Blacksburg (Table 3). The university also has a presence in northern Virginia with several facilities located around the greater Washington, D.C., metro area, and has a small presence in Roanoke and Richmond, Virginia. Undergraduate and graduate students made up 64% and 36% of the sample, respectively. Among the undergraduate participants, the distribution of freshmen, sophomores, juniors, and seniors was approximately even. In addition, 12% of the undergraduate students identified as being transfer students. Among the graduate participants, more doctoral students (58%) than master's students (42%) responded to the survey. Some 14% of undergraduates had a GPA between 2.5–3.0, 34% between 3.0–3.5, and 47% between 3.5–4.0. In contrast, the majority of (87%) of graduates had a GPA between 3.51–4.00. Around two-thirds (66%) of respondents were in-state students, 82% spoke English as their first language, 20% were first-generation college students, 8% indicated they had a disability, and 12% were married. The majority of respondents were white or Caucasian (70.3%). Table 3 also shows how these variables align with university data.

**Table 3. Comparison of the sample and survey population.**

| Variable | Sample | | University | |
|---|---|---|---|---|
| | % | N | % | N |
| **Level of education** | - | **2,112** | - | **35,483** |
| Undergraduate | 63.9 | 1,343 | 80.6 | 28,593 |
| Graduate (includes Professional student) | 36.1 | 759 | 19.4 | 6,890 |
| **Campus location** | - | **2,105** | - | **21,139** |
| Blacksburg | 93.5 | 1,968 | 96.5 | 20,395 |
| Roanoke | 1.6 | 34 | 0.4 | 91 |
| Northern Virginia | 4.3 | 91 | 2.9 | 611 |
| Richmond | 0.2 | 5 | 0.2 | 42 |
| Virginia Beach | 0.3 | 6 | - | - |
| Newport News | 0.1 | 1 | - | - |
| **Academic level (undergraduate)** | - | **1,339** | - | **28,480** |
| First-Year/Freshman | 24.5 | 328 | 12.5 | 3,566 |
| Sophomore | 24.7 | 330 | 21.6 | 6,165 |
| Junior | 24.5 | 328 | 24.8 | 7,056 |
| Senior | 26.4 | 353 | 41.1 | 11,693 |
| **Transfer student (undergraduate)** | - | **1,343** | - | **28,593** |
| Yes | 11.9 | 160 | 10.5 | 3,016 |
| No | 88.1 | 1,183 | 89.5 | 25,577 |
| **Academic level (graduate)** | **100** | **759** | | **6,219** |
| Doctoral level | 58.0 | 440 | 47 | 2,923 |
| Master level | 42.0 | 319 | 53 | 3,296 |
| **GPA (undergraduate)** | - | **1,288** | - | **17,030***  |
| <2.0 | 0.7 | 9 | 2.9 | 501 |
| 2.00–2.50 | 4.4 | 57 | 6.6 | 1,126 |
| 2.51–3.00 | 13.7 | 176 | 16.9 | 2,871 |
| 3.01–3.50 | 34.2 | 441 | 35.5 | 6,052 |
| 3.51–4.00 | 47.0 | 605 | 38.1 | 6,480 |
| **GPA (graduate)** | - | **673** | - | **5,822***  |
| <3.0 | 1.5 | 10 | 4.8 | 282 |
| 3.01–3.50 | 11.3 | 76 | 18.5 | 1,076 |
| 3.51–4.00 | 87.2 | 587 | 76.7 | 4,464 |
| **In-state & Out-of-state (undergraduate)** | - | **1340** | - | **28,593** |
| In-state | 73.7 | 988 | 71.2 | 20,353 |
| Out-of-state: domestic | 21.1 | 283 | 23.5 | 6,707 |
| Out-of-state: international | 5.1 | 69 | 5.4 | 1,533 |
| **In-state & Out-of-state (graduate)** | - | **766** | - | **6,890** |
| In-state | 51.3 | 393 | - | 2,998 |
| Out-of-state: domestic | 21.0 | 161 | 30.7 | 2,113 |
| Out-of-state: international | 27.7 | 212 | 25.8 | 1,779 |
| **First-generation college student** | - | **2,058** | - | **35,483** |
| Yes | 20.2 | 416 | 14.2 | 5,038 |
| No | 79.8 | 1,642 | 85.8 | 30,445 |
| **Disability** | - | **2,070** | - | **35,483** |
| Yes | 7.7 | 159 | 10.5 | 3,725 |
| Maybe | 6.1 | 126 | - | - |
| No | 86.2 | 1,785 | 89.5 | 31,758 |

*(Continued)*

**Table 3.** (Continued)

| Variable | Sample | | University | |
|---|---|---|---|---|
| **Race/Ethnicity** | | **2096** | - | **34,486**** |
| American Indian or Alaskan Native | 1.4 | 30 | 0.1 | 42 |
| Asian | 20.5 | 429 | 10.4 | 3,583 |
| Black or African American | 5.8 | 121 | 5.1 | 1,749 |
| Caribbean | 0.7 | 15 | - | - |
| Hispanic or Latino | 7.2 | 151 | 7.4 | 2,566 |
| Middle Eastern or North African | 3.2 | 67 | - | - |
| Native Hawaiian or Pacific Islander | 0.5 | 11 | 0.1 | 34 |
| Sub-Saharan African | 0.6 | 13 | | - |
| White or Caucasian | 70.3 | 1473 | 62.1 | 21,429 |
| Self-identify as another race/ethnicity | 1.0 | 21 | 14.7 | 5,083*** |
| Identify as multiple | 9.9 | 208 | - | - |

Notes:

* Excludes 0 and missing data.

** 997 students did not report their race/ethnicity. The university data does not include the following categories: 'Caribbean,' 'Middle Eastern or North African,' and 'Sub-Saharan African.'

*** Self-identify as another race/ethnicity includes 'Two or more races' (1,627) and 'Nonresident Alien' (3,456).

In addition to the USDA instrument (Table 1) and the key sample variables (Table 3), the survey asked additional questions relating to food access, mental health, and housing. It also included additional questions relating to respondent characteristics such as gender identity, sexual orientation, marital status, and income. The mental health questions were based on the McGill Quality of Life Questionnaire [52,53]. The housing questions were based on a screening question from the U.S. Centers for Medicare and Medicaid Services (CMS) Health-Related Social Needs Screening Tool [54] and from a study by Goldrick-Rab, et al. [55] on food and housing security.

## Results

Approximately 16% of undergraduate and graduate students can be classified as having low food security (Table 4). Masters students had a slight lower level of food insecurity (14.7%) than doctoral students (17.7%), but this difference was not statistically significant.

The socio-demographic and economic characteristics linked to a higher likelihood of experiencing low food security included:

- having a GPA of less than 3.0

- having a disability

- being an international student

**Table 4. Food security by student status.**

| | High food security (%) | Low food security (%) |
|---|---|---|
| **Undergraduate** | 84.1 (n = 1130) | 15.9 (n = 213) |
| **Graduate** | 83.5 (n = 634) | 16.5 (n = 125) |
| Master level | 85.3 (n = 272) | 14.7 (n = 47) |
| Doctoral level | 82.3 (n = 362) | 17.7 (n = 78) |

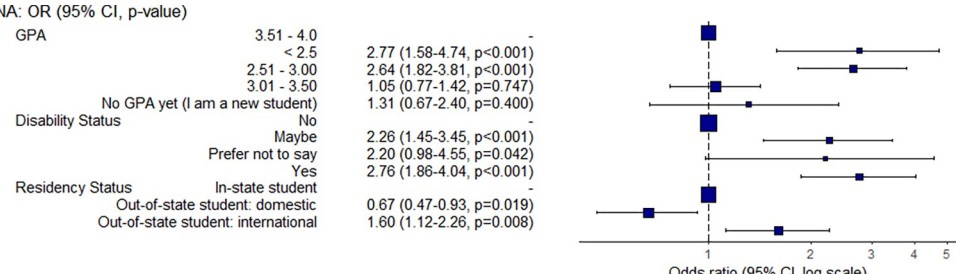

**Fig 1. Odds ratio and confidence interval for GPA, disability, and residency variables.**

- being a first-generation student

- being a transfer student

- going into debt to pay for food

- being a Black or African American student

- having poor mental health

- having uncertain living arrangements, and

- having no medical insurance.

Students with a lower GPA were more likely to have a low food security status when compared with students with higher GPAs (Fig 1). For example, students with a GPA of less than 3.0 were around 2.6 times more likely to have a low food security status than students with a GPA of between 3.5 to 4.0. The relationship between academic performance and food security is complex and often accompanied by other factors such as having a low income and needing to work and/or experiencing poor mental health. As one student explained:

> "*Even with free vouchers from work, and having cut back on my eating habits, I still don't have enough money on my dining plan to eat anymore. . . . The stress of having to work just to afford tuition has driven my grades through the floor, and now I'm genuinely concerned that, because of my mental problems due to COVID, and because of my constant exhaustion on the days I work, that I'm not going to be accepted back . . . next semester. I would love to get counseling, but everything they're doing is virtual, and constantly being stuck staring at a screen is half my problem*!"

Students who self-assessed that they had a disability were 2.8 times more likely to have a low food security status when compared with students without a disability (Fig 1). The type of disability was not requested in order to maintain student privacy. One student provided some insight on how their disability impacts their food access:

> "*Having my disability means it takes about 3 times as long to make a meal and about twice as long to shop. Thus, I lost most of my money to spoiled food I didn't have time to cook. Or I just ran out of time to shop for food. Fast food is/was not an option either. It should never be food vs. school, but it was until I was accepted into the food assistance program.*"

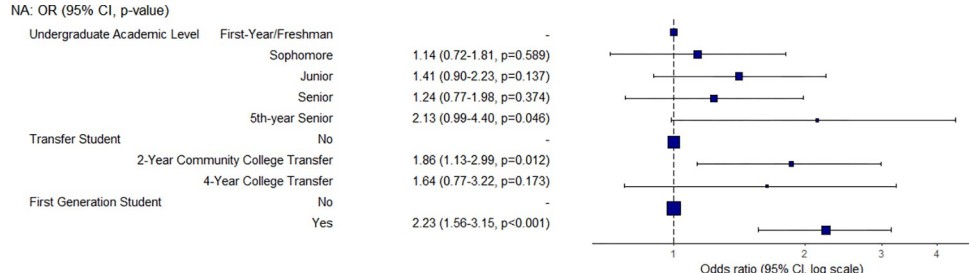

**Fig 2. Odds ratio and confidence interval for undergraduate academic level, transfer, and first-generation variables.**

International students were 1.6 times more likely to have a low food security than in-state students. In contrast, out-of-state domestic students were 33% less likely to be food insecure than in-state students.

First-generation students were 2.2 times more likely to have a low food security status than second-generation students, and transfer students from a 2-year community college were 1.9 times more likely than non-transfer students (Fig 2). When comparing the different years of undergraduate students, freshmen appear to be the most food secure, but the differences between years is not significant.

There is no significant difference between the food security of master's and doctoral students (Fig 3 and Table 4). However, graduate students with a 20-hour (full) or 10-hour (half) assistantship were 3.0 and 3.6 times, respectively, more likely to have a low food security status when compared with graduate students with no assistantship. The comments below provide some insight into the financial challenges facing graduate students.

"*The stipend we get is just enough to live on. I am constantly worried about money and would not have any financial resources of my own if a major event happened (health emergency, car troubles, etc.).*"

"*The graduate comprehensive fees I have to pay on top of taxes make my income very limited as an international student on [an . . .] assistantship. I am also not allowed to seek additional employment under the terms of my visa so it is a huge difficulty to manage expenses with rent and utilities.*"

"*My wife will be joining me as a dependent in May. I'm afraid that my assistantship stipend will not be sufficient for us to live a healthy life. We will also be not able to have a child until I graduate because it would be financially impossible to support my wife and a child with this stipend amount.*"

Students who reported building up debt to pay for food or having spent less on other budget items to pay for food were significantly more likely to experience low food security (Fig 4).

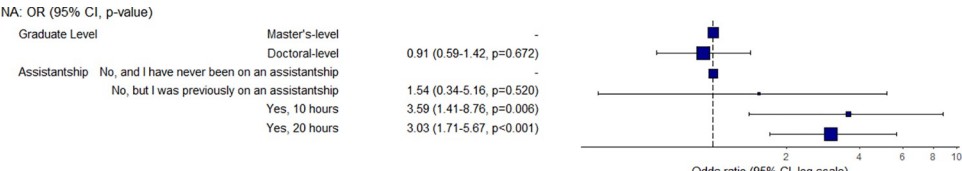

**Fig 3. Odds ratio and confidence interval for graduate level and assistantship variables.**

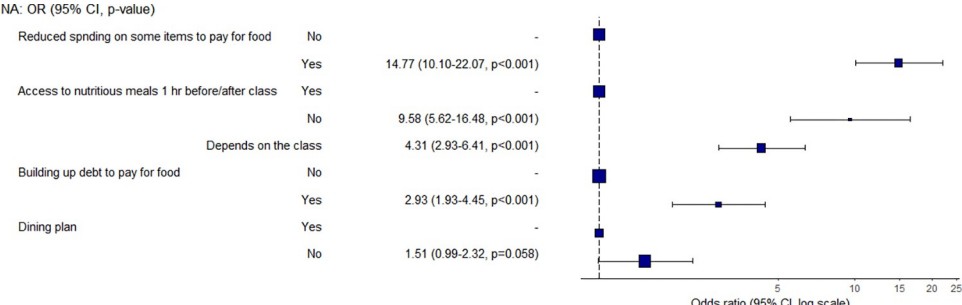

**Fig 4. Odds ratio and confidence interval for food- and expenditure-related variables.**

Those students who were building up debt to pay for food were 2.9 times more likely to have a low food security status than students who were not, whereas those who were spending less on other budget items when compared with those who were not were 14.8 times more likely to have low food security. Importantly, students also reported their lack of access to food was negatively impacting their ability to enjoy the full university experience.

Students on the main campus with access to a dining plan were less likely to have food access challenges, although students with no access to food one hour before/after class were 9.6 times more likely to have a low food security status than those with access. Further, having a dining plan does not mean students have access to the food they need due to dietary restrictions.

> "*I really hope campus dining will one day become more inclusive to people with restricted diets. I felt estranged from my peers who were gaining weight from campus food while I was losing weight as a result of it.*"

With regard to race/ethnicity, the likelihood of experiencing low food security increase by 2.3 times if a student identifies as being Black or African American (Fig 5). While none of the other race/ethnicity groups have a statistically significant likelihood of experiencing food insecurity, Sub-Saharan African and Hispanic or Latino students are groups that may be at risk. In contrast, White or Caucasian students are the only group to have a reduced likelihood of experiencing low food security, but this finding is not significant.

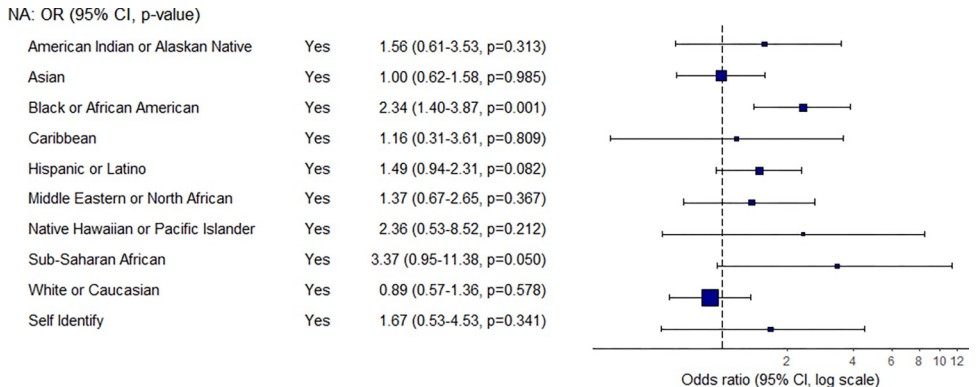

**Fig 5. Odds ratio and confidence interval for race/ethnicity.**

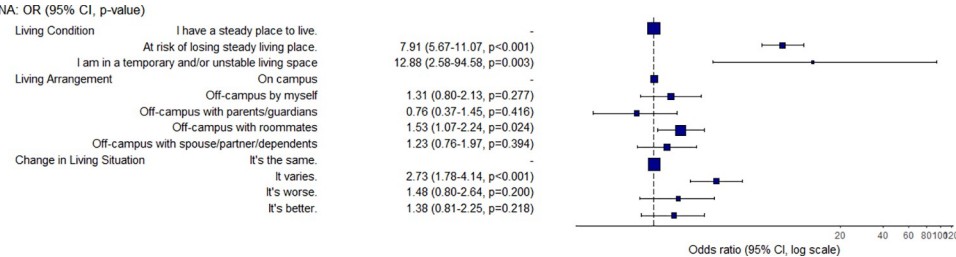

**Fig 6. Odds ratio and confidence interval for housing variables.**

Housing and food security are often intertwined. The results reveal how housing and a student's living situation have significant associations with their food security status (Fig 6). Students who live off-campus with roommates were 1.5 times more likely to have a low food security status when compared with students living on campus. Further, students who were worried about losing their current living situation or were living in a temporary or unstable space were 7.9 and 12.9 times, respectively, more likely to have a low food security status when compared with students who have a steady place to live. For comparison, around 12% of students who had a steady place to live experienced a low level of food security, compared with 55% of students who had a steady place to live, but were worried about losing it (see S1 Table). Further, four of the six students (66.7%) who reported their living situation was worse or had varied in the last six months were found to have a low food security status. The need to strategize on how to meet basic needs by balancing access to housing and food was a theme in the written responses. As one student explained:

> "When I was couch surfing and living with my partner, food was about 80% of where my income went. The last time I was on campus, it was common to hear students on dining plans talking about running out of meal credit, surviving off of peanut butter, and/or strategizing about the cheapest places to eat to maximize plans. I would personally go to events and find places with free food since I couldn't always afford decent food."

For undergraduate students in particular, there is a difference in food accessibility when living on-campus versus off-campus. Around six percent of respondents were living with their parents or guardians. This group had one of the highest levels of food security (87.2%), but 13% of students were still experiencing food access problems (see S1 Table). Approximately 12% of the students living on campus had a low level of food security, compared with 17% of students living off campus by themselves or with roommates. In this case, food security may be most related to the ability to access and prepare nutritious foods. One student shared:

> "My parents would pay for my dining plan and I was eating fairly healthy food. I still live . . . with my roommates [off campus], but I am now paying for my own food. I usually eat fast food now and it has greatly affected my quality of life."

Mental health was also strongly associated with food access and security. Students who reported being depressed or having limited control over their life were around 2 or 6 times, respectively, more likely to have a low food security status when compared with students who reported no concerns (Fig 7). However, when asked to consider all parts of their life over the prior week, students who reported having a poor or terrible quality of life were 11.2 and 27.9

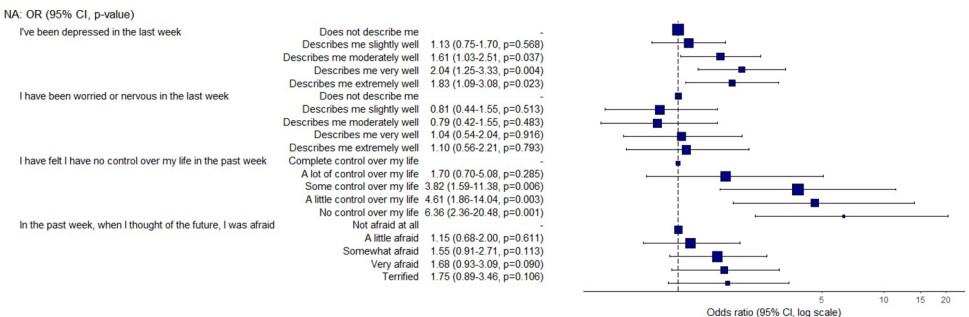

**Fig 7. Odds ratio and confidence interval for mental health variables.**

times, respectively, more likely to be food insecure than students who described their life as excellent (Fig 8). With regards to weight, students who reported losing weight were found to be 2.3 times more likely to have a low food security status than students who reported no significant weight change. Finally, students without health insurance were found to be 2.6 times more likely to have a low food security status than students with health insurance.

Given the ongoing COVID-19 pandemic, students were asked how their lives have been impacted (Fig 9 and S1 Table). Students who experienced a decline in their disposable income were 3.3 times more likely to have a low food security status than students who did not select this option. In contrast, students who reported an increase in their disposable income were the only group to experience a reduction in the likelihood of being food insecure, but this result is not significant. Other factors associated with an increased likelihood of experiencing low food security during the pandemic were the loss of access to regular childcare or transportation, a reduction in physical health, the loss of a job, and a decline in housing security.

A number of respondents were benefiting from food assistance programs, including a new service run by the university that provides participants with weekly bags of produce and other essential items. Two respondents who benefited from the university's service provided the following feedback:

> "*The Market . . . has made a huge difference along with finding a friend who cooks meals easily. I'm finally eating 3 meals a day.*"

> "*I am so grateful for local resources . . . including The Market . . .. School is more stressful than ever, but I guess that's just the way junior year in chemical engineering goes (while enduring a pandemic). I'm taking one day at a time dealing with school stresses, but I am so thankful I don't have to worry about my next meal!*"

However, one student highlighted a challenge with food services that are based on a more limited set of food options:

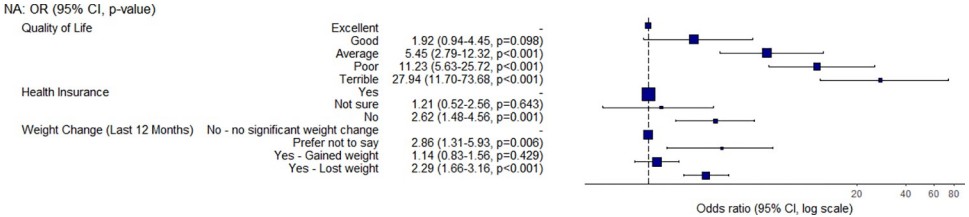

**Fig 8. Odds ratio and confidence interval for quality of life, insurance, and weight variables.**

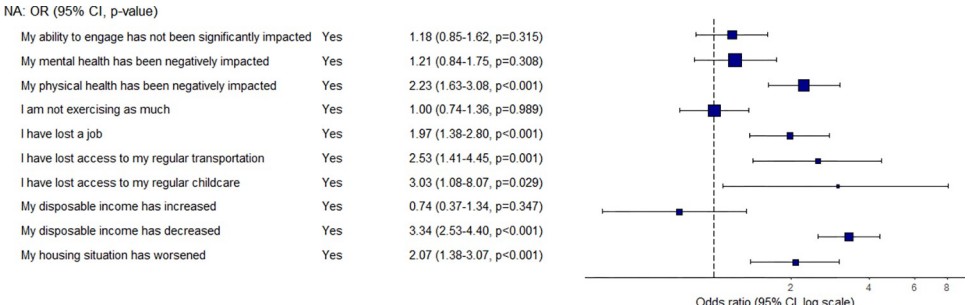

**Fig 9. Odds ratio and confidence interval for COVID-related question.**

"*Programs like the market or other food pantries would be more effective if they gave out gift certificates to local stores. This would improve local spending and support these businesses, and students using them could purchase what they wanted instead of relying on boxed or canned foods, or foods they don't know what to do with or don't normally eat.*"

## Discussion and recommendations

This section discusses the main findings from this study and presents a series of recommendations that are intended to assist institutions of higher education in enhancing the food security of their students.

### Reaching students at risk of being food insecure

Growing evidence reveals how institutions of higher education can use student characteristics, experiences, and their access to financial resources to identify students at risk and develop processes and resources to ensure food security [12,17]. Student characteristics that were found to be associated with a higher likelihood of food insecurity include being a first-generation or transfer student, having a disability, having a GPA of less than 3.0, and identifying as a Black or African American student.

First-generation college students have been shown to be more likely to experience food insecurity than their peers in this study and others [56,57]. A first-generation student often faces barriers in accessing resources because they do not know what resources exist and cannot ask a parent for guidance [12]. Similarly, transfer students may be unfamiliar with a university's support systems and who to ask for assistance.

Some researchers recommend the screening of first-year students to identify individuals at risk and to develop policy and programmatic initiatives to support their college experience [33]. While likely effective for incoming students, there is evidence that food security status tends to be 'fluid' and impacted by irregular access to financial resources and unexpected expenses relating to healthcare, vehicle repairs, etc. [20]. Thus, other mechanisms may be needed to reach existing students.

Having a disability is one of several factors that were found to increase the likelihood of experiencing food insecurity. The results reveal the additional time students with a disability may need to purchase food and make meals, which can reduce the time available for study.

Students with a low GPA were found to be more likely to be food insecure than students with a high GPA. Poor nutrition can lead to diminished cognitive performance [58,59]. Additionally, the stress of procuring the resources needed to access food can also lead to reduced academic performance [10,31]. This stress can lead to further negative emotional wellbeing, as was evident in the written responses provided by students.

Finally, numerous studies, including this one, reveal that students identifying as belonging to a racial or ethnic minority have a higher likelihood of experiencing low food security [9,30].

**Recommendations.** Developing strategies to inform first-year and existing students, especially those at higher risk of food insecurity (i.e., students with a disability, minority students, and low academically performing students), of the resources available to them throughout their studies is recommended. University administrations can work most effectively through student groups, associations, and initiatives that already reach students most at risk of experiencing food security to provide tailored information on available resources. Primarily, these groups and initiatives can be used to inform students of resources available to support them during times of food scarcity. They can also be used to obtain feedback on the underlying factors that contribute to an increased likelihood of experiencing food insecurity, which can enable academic institutions to tackle the underlying causes of food insecurity.

## Meeting dietary needs

A common experience amongst students, no matter their food security status, is difficulty accessing food on campus that meets a specific dietary need (e.g., allergy, religious requirements). While local resources such as community gardens can provide culturally-relevant food, the start-up costs and competitiveness to access such spaces can be prohibitive.

**Recommendations.** A first step to ensuring that students have access to food that meet their dietary requirements is to review the foods available on campus through dining halls and other outlets. This information can then be used to target interventions aimed at enhancing the diversity of food offerings that serve a range of dietary needs.

## Housing security

Housing security is strongly linked with food security [2,12]. As the average rent continues to increase with the rise of enrollments at the university, the financial burden to students will also rise; potentially giving way to an increased risk of food insecurity. The survey revealed that students who were worried about losing their housing were more likely to be experiencing low food security than students living in a steady place. For the minority of students who were living in a temporary/unstable place, this number increased significantly. Thus, there is a clear correlation between housing and food security, and it is likely that students experiencing housing insecurity are also likely to have food access challenges.

**Recommendations.** In addition to providing enhanced compensation for graduate students, providing more affordable housing options on campus and in the community would help support students facing housing insecurity.

## Mental health

The stress of providing for basic needs while trying to engage in education takes an intensive toll on the mental health of students. The association between food, financial, or housing insecurity and mental well-being has been well documented [11,15,32,33]. About one-third of students who reported that in the prior week they were depressed, nervous/worried, afraid of the future, or had no control over their life were also experiencing low food security. Student feedback revealed a range of factors, including but not limited to food access, that were impacting mental health. Many students spoke about the negative impact of COVID-19 restrictions, challenges they faced engaging with online courses, and more broadly of concerns about the future. Even before the pandemic, the university had experienced an increase in demand for mental health services [60], which were put under further pressure during the pandemic.

**Recommendations.**  Services that provide mental health support to students can also be an entry point for connecting students in need with resources to increase food access. In response to increased demand for mental health services, the university hired several new counselors and placed them in colleges and on different campuses in an effort to enhance student access to mental health services. The following year, students were given access to the TimeCare App which provides a range of virtual health services, including immediate or scheduled counselor sessions and health coaching. The latter service includes information about nutrition, highlighting an opportunity to directly connect the mental health and food access services available to students. Finally, encouraging students to enroll in available services before they experience a mental health problem can help ensure they receive the help they need when they need it most.

## Financial security

Having sufficient funds to pay for food was found to be one of the most critical factors related to low food security. For example, nearly one half of students who were building up debt (e.g., on a credit card) to pay for food or were spending less on other budget items (such as heating/cooling) to pay for food were food insecure. It is likely that the COVID-19 pandemic put additional financial burdens on students as their families were impacted by the economy-wide shutdowns. Further, one fifth of graduate students who were receiving a full (20-hour) assistantship were experiencing low food security.

In response to these types of concerns, in 2022 the university passed a Resolution for Equitable and Cost-of-Living-Responsive Graduate Student Compensation [61] that commits the institution to establishing a strategic goal and funding plan for the provision of equitable graduate assistantships. An important aspect of this resolution is the focus given to meeting the needs of "international and historically marginalized students," who are among those groups most at risk of food insecurity.

**Recommendations.**  Providing dinning plan subsidies is an effective way to increase food access, without revealing to others using the services that a student is receiving assistance. Another way to increase financial support for students would be to expand employment opportunities on campus such as through the Federal Work-Study (FWS) program. The FWS provides institutions of higher education with funds to hire students in financial need on an hourly wage during the academic year. Students can work for university colleges, departments, or programs, or off-campus with approved employers and non-profit organizations. Participating institutions need to apply each year for FWS funding and be able to cover up to 50 percent of a student's hourly wage. Finally, identifying opportunities to provide affordable housing for students would be an effective way to reduce the cost of living and free up financial resources for food and other critical services.

## Food access services

The percentage of survey respondents found to have a low food security status (16%) is lower than what is typically found in studies at other comparable institutions of higher education [9,30,38]. Following an initial survey of food access and security at the university [62], a food security taskforce was convened in 2020 that identified a series of barriers students face when accessing food and presented a range of recommendations and actions to address them [63]. The most significant action taken was the creation of a free food service that could provide over 100 students a week with access to seasonal produce and other food items. This service began as a pilot program in 2020 and was formally launched in 2022 with additional donor support. In parallel with this, the university's Dean of Students Office dedicated a portion of its

emergency assistance fund to support students with food access needs via one-time grants of typically between $600-$800.

Since 2019, the university has also significantly enhanced its communication of these support services, which also includes information about off-campus resources such as a local food pantry that is available to all students. In addition to the university's actions, the Coronavirus Aid, Relief, and Economic Security Act (CARES Act) provided three rounds of direct relief payments to eligible adults and qualifying children, two of which were released during the year prior to the survey.

While it is not possible to make a direct causal connection between the higher level of food security and the actions taken by the university or funding that may have been received via the CARES Act, written feedback from participants indicates that a number of students benefitted significantly from the university support they received. The higher level of food security could also be related to a self-selection bias due to COVID stressors. For example, students experiencing a decline in mental and physical health may not have had the capacity to complete the survey.

**Recommendations.** Food access considerations permeate a wide range of university functions, from decisions about what food to offer in dining halls and how to structure affordable dining plans, to the location of food services and what services to provide at these locations, to considering how the scheduling of courses may impact a student's access to food services, to how to reach all students who need support when they need it, to the creative design of the support services that are provided. It is also critical to connect mental health, housing, and food access programs to ensure that all university services are in sync and students are provided with comprehensive support. Further, efforts should be taken to ensure that any transportation barriers students might face in accessing these services are addressed [12]. In addition to sharing information with students about the services available to them through emails, posters, etc., other strategies including listing these services on all syllabi to reinforce the link between overall health and academic performance.

## Reducing survey fatigue and burden

Repeated surveying of student bodies to ascertain food security can be difficult, create respondent burden, and deals with complex and sensitive subjects.

**Recommendations.** Developing systems and processes that can identify students at risk of experiencing food insecurity, without the need for repeated surveys, will be a critical step in tackling this issue at college campuses across the US. Integrating food security considerations into student advising systems/software may be an effective way for advisors to identify students at risk and connect them with available resources.

## Limitations and direction for future research

As with all cross-sectional research, the results from this study should be interpreted by considering its potential limitations. While all students at the university were sent the survey, only 6% responded. Table 3 indicates that graduate students were overrepresented (36% sample vs. 19% university) and undergraduate students were underrepresented (64% sample vs. 81% university). Within the undergraduate respondents, freshmen were over overrepresented (25% sample vs. 13% university) and seniors were underrepresented (26% sample vs. 41% university). First-generation students we also overrepresented (20% sample vs. 14% university). With regards to race/ethnicity, white or Caucasian (70% sample vs. 62% university), Asian (21% sample vs. 10% university), Native American or Alaskan (1.4% sample vs. 0.1% university), and Native Hawaiian or Pacific Islander (0.5% sample vs. 0.1% university) students were

overrepresented. However, the sample was more representative with regards to student GPA, transfer students, in-state vs. out-of-state students, and black or African American and Hispanic or Latino students.

The cross-sectional research design only permits the examination of associations between food security and other key variables of interest, rather than the establishment of causal links. Hence, there is a need for longitudinal studies that determine the mechanisms that cause students to transition between different levels of food security. Further, the USDA food security instrument relies on a 12-month recall period. Since data collection occurred during the end of the spring 2021 semester, students may have been referring to their prior summer experience when responding to the survey. While only one student referenced how they used their summer income to support their studies, it is not clear whether students considered the entire 12-month period or focused on the academic semesters. While there is a version of the USDA instrument with a 30-day recall period, this is intended to study the impacts of a specific program. Given that these two versions of the USDA instrument have limitations, it may be interesting to test a revised version of the instrument that does not refer to 'households' and has a recall period that matches the academic calendar.

While this research did not capture the type of disability a student might have in order to maintain student privacy, understanding the factors associated with student disability status and food insecurity is an important avenue for future research [64].

Finally, since this study focuses on a single university, care should be taken when trying to generalize the findings to other institutions of higher education.

## Conclusions

This mixed methods study identifies the socio-demographic and economic characteristics that were linked to a higher likelihood of students experiencing low food security at a large southeastern land grant university. The findings reinforce those of comparable studies and reveal the interconnected relationship between food and housing security and mental health. The study concludes by identifying a series of recommendations that universities in the U.S. can consider to ensure that students have access to the resources they need to be successful while pursuing their degrees.

## Supporting information

**S1 Table. Food security by selected indicators.** This table presents the USDA food security metric (i.e., high versus low food security) for each of the main variables included in the study. (DOCX)

## Acknowledgments

We would like to thank Bo Guan and Daniel Kim in the Chris North research group at Virginia Tech for the assistance they provided in analyzing the data. We would also like to thank Nikki Lewis for guidance on the data visualization.

## Author Contributions

**Conceptualization:** Ralph P. Hall, Jessica Agnew.

**Formal analysis:** Ralph P. Hall, Jessica Agnew, Wei Liu, Lana Petrie, Chris North.

**Funding acquisition:** Ralph P. Hall, Jessica Agnew.

**Investigation:** Ralph P. Hall, Jessica Agnew.

**Methodology:** Ralph P. Hall, Jessica Agnew.

**Validation:** Wei Liu, Chris North.

**Visualization:** Wei Liu.

**Writing – original draft:** Ralph P. Hall, Jessica Agnew, Wei Liu, Lana Petrie.

**Writing – review & editing:** Chris North.

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
