## [Decision Letter · Decision Letter 0]

29 May 2023

PONE-D-23-05009Systematic investigation of inadequate food access at a large southeastern land grant universityPLOS ONE

Dear Dr. Hall,

Thank you for submitting your manuscript to PLOS ONE. After careful consideration, we feel that it has merit but does not fully meet PLOS ONE’s publication criteria as it currently stands. Therefore, we invite you to submit a revised version of the manuscript that addresses the points raised during the review process.

We look forward to receiving your revised manuscript.

Kind regards,

Jocelyn Turner-Musa

Academic Editor

PLOS ONE

4. Thank you for stating the following in the Funding Section of your manuscript:

“The initial research that made this study possible was supported by funding from the Virginia Tech Center for International Research, Education, and Development (CIRED), the office of Outreach and International Affairs (OIA), and the College of Science.”

Additional Editor Comments:

The authors are commended for their attempt to add to the extant literature on the topic of food accessibility in institutions of higher education.  The utilization of a mixed methods approach to capture both quantitative and qualitative data on the topic is laudable.  However, as pointed out in the review, there are several areas in need of revision.

In the Introduction the inclusion of supporting references is needed.  For instance, on lines 39 – 44 the authors fail to provide empirical support for statements regarding evidence of the role of food insecurity on student performance and quality of life.  On line 49 it is unclear as to what “rite of passage’ refers to.  As Reviewer 1 queries “Is this referring to experiencing challenges accessing food as a “rite of passage.” This statement is vague and should be clarified.  The authors should also check lines 55 – 56 (i.e., “There are is..”) and line 67 (i.e., Recent research explores…) for grammatical errors.  More information is needed on the 10-item USDA Household Food Security Instrument.  This is the instrument used to assess food security and more details are needed on how the measure is used and how it has been validated if not on a college student population as mentioned.  This measure assesses the primary construct of interest and should be fully described. 

In the Methods section on line 132 it states that “The word household was removed from the module” however, in Table 1 the word “household” is included.  Please clarify.  It is also unclear if Question 1 in the table is a screener question.  It appears to be a screener question and if so, this should be stated.  A description of all measures assessed should be included in this section.  For example, demographic questions pertaining to race/ethnicity and income are not included.  There is also no mention of a measure or items that assess mental health. On line 167 the qualitative method used to analyze open-ended responses is provided.  More information on this method and a reference is needed.  What prompts or questions were provided to participants?  Was inter-rater reliability assessed? Was qualitative software used to identify themes?  On line 165 it is mentioned that “coding was conducted using Word”, this is vague. 

The Results section could be strengthened if more details about the sample and measures are provided.  Without more details (e.g., race/ethnicity) it is difficult to interpret the findings. What percentage of the sample is Black or African American?  Has sample size been adjusted for small n’s?  The figures are blurred and difficult to see.  Be sure to follow submission guidelines for submitting figures and graphs.  The authors provide recommendations at the end of each result presented.  It may be best to just present the findings in the Results section and in the Discussion section provide a summary of the findings with recommendations.  

In the Discussion section, qualitative responses are included.  It is more appropriate to present these findings in the Results section as this is a mixed-methods study.  As cited earlier, more details are needed on the qualitative component of the study and themes that emerged.   

Other comments relate to other areas the authors should consider.  As the reviewer suggests, provide potential explanations to why the prevalence of food insecurity was lower in this study than in others. Knowing more about the university and campus locations students were recruited from may assist with this. How many of these students live on campus versus off campus?  How many live with their family or a roommate as this too may aid in data interpretation.  Additionally, there seems to be vacillation between the term “low food security” and “food insecurity”.  Are the authors suggesting a difference between the two terms?  Are they the same?  If so, it may be easier and more efficient to use one or the other.  An operational definition of food insecurity might help.  Given that the study was conducted in April 2021 during the Covid-19 pandemic, this should also be considered as a potential factor influencing findings.  In addition to participant recall, this too may be a potential confound and limitation.

Overall, the topic is very interesting and has the potential to address factors associated with food insecurity on a college campus. As written, the paper is not accepted for publication but with significant revisions will be reconsidered.

Reviewers' comments:

Reviewer's Responses to Questions

**Comments to the Author**

1. Is the manuscript technically sound, and do the data support the conclusions?

Reviewer #1: Yes

2. Has the statistical analysis been performed appropriately and rigorously? 

Reviewer #1: Yes

3. Have the authors made all data underlying the findings in their manuscript fully available?

Reviewer #1: Yes

4. Is the manuscript presented in an intelligible fashion and written in standard English?

Reviewer #1: Yes

5. Review Comments to the Author

Reviewer #1: Introduction

Lines 39-44: Add references

Lines 48-50: The wording of this sentence is a little unclear. It sounds like it’s saying that access to food is considered a “rite of passage.” Is this referring to experiencing challenges accessing food as a “rite of passage.”

Lines 55-56: This is the first time the 10-item USDA Household Food Security Survey instrument is mentioned. It would be helpful to briefly mention what this is and that it’s a commonly used measure for assessing food security status in the US.

Lines 58-60: How does it allow for comparable results?

Methods

Line 132: This says the word household was removed from the module, but the first question in table 1 uses household

Since “high” and “low” are two of four USDA categories, I suggest referring to the two groups as food secure and food insecure as commonly done in other studies. The results section is confusing because it’s unclear if you’re referring to just the low food security group or low/very low combined. There is also inconsistency in the results section as to how it’s being referred to.

The methods section only describes the questions used to assess food security status. Briefly describe the other questions on the survey. There doesn’t need to be as much detail as for food security but some mention of other information being collected would be helpful.

Lines 159-160: Was R used just for the odds ratios or also for the chi-square and Fisher’s exact tests described in the next paragraph. If it was used for all, you may want to move that sentence to the beginning or end of the information on statistical analyses to help clarify.

Lines 167-169: How many coders were there and were responses double-coded or coded by a single coder?

Results

Lines 175 and 176: Be consistent in use of commas in numbers.

Line 176: You may want to provide a little context about the university in the methods section as this is the first time having a main campus was mentioned. Many universities only have a single campus, so you may want to mention how many campuses there were in the methods and if the majority of students attended the main campus.

Table 4: Rather than saying “by Selected Demographic Characteristics” you may want to say something like student classification.

Line 220: The tables and figure seem to indicate that the question on being a first generation college student was a yes/no question and that data wasn’t collected to determine if a student was a second generation college student. Not all students who are not first generation college students are second generation – some may have grandparents, great grandparents, etc. who also attended college.

Lines 235-236: I’m not sure that saying there is an important difference between the groups is the best way to reference that there was a large difference in the odds ratios. You may want to take out this sentence or reword to focus more on there being a difference in size of odds rations.

Discussion

Line 387: The way this is worded sounds like 7% of those who lived with their parents. Is this trying to say that 7% of students in the sample lived with their parents?

Lines 475-476: You may want to briefly explain what work study is.

Lines 480-515: There are many reasons why the prevalence of food insecurity was lower in this study than some others. You may want to mention some potential other reasons as well. This is a great example of a program the university is offering, but will a program reaching up to 115 students from a university of over 35,000 students likely be having a major impact on the prevalence of food insecurity on the campus?

Line 559: For future studies, you may want to keep in mind that there is both a 12-month and 30-day recall period for the 10-item US Household Food Security Survey Module.

Limitations

Another limitation to mention is external generalizability as this study included students from a single university

6. PLOS authors have the option to publish the peer review history of their article (what does this mean?). If published, this will include your full peer review and any attached files.

Reviewer #1: No

---

## [Author Response · Author response to Decision Letter 0]

12 Nov 2023

• RESPONSE: Thank you for providing these guidelines. The manuscript has been reformatted so that it complies with PLOS ONE’s style requirements.

• RESPONSE: As indicated in the methods section, the survey targeted undergraduate and graduate students who were enrolled either as part- or full-time students and aged over 18. Students who did not meet these criteria were excluded. We now explain that consent was obtained at the start of the online survey once the participant had read the consent script and had selected “yes” to a question asking whether they agreed to participate in the study.

• RESPONSE: We have included our R code along with the research protocol, survey instrument, and cleaned data set on our OSF site. This link will be made public if the paper is accepted: https://osf.io/bnuqv/?view_only=977d9eefa53d4e61b0ce61eada8cb048

4. Thank you for stating the following in the Funding Section of your manuscript:

“The initial research that made this study possible was supported by funding from the Virginia Tech Center for International Research, Education, and Development (CIRED), the office of Outreach and International Affairs (OIA), and the College of Science.”

• RESPONSE: We have removed the funding statement from the manuscript as requested and will revise the Funding Statement as follows: “The initial research that made this study possible was supported by funding from the Virginia Tech Center for International Research, Education, and Development (CIRED), the office of Outreach and International Affairs (OIA), and the College of Science. These entities had no role in the study design, data collection and analysis, decision to publish, or preparation of the manuscript.”

• RESPONSE: If the manuscript is accepted for publication, all of the data used in the analysis will be publicly accessible via our OSF site: https://osf.io/bnuqv/?view_only=977d9eefa53d4e61b0ce61eada8cb048. We can provide the DOI for the data set as soon as the OSF site is made public. 

• RESPONSE: The Appendix has been renamed “S1 Table.” All in-text citations have been changed to “S1 Table,” and the following caption has been added at the end of the paper: “S1 Table: Food Security by Selected Indicators. This table presents the USDA food security metric (i.e., high versus low food security) for each of the main variables included in the study.”

Additional Editor Comments:

The authors are commended for their attempt to add to the extant literature on the topic of food accessibility in institutions of higher education. The utilization of a mixed methods approach to capture both quantitative and qualitative data on the topic is laudable. However, as pointed out in the review, there are several areas in need of revision.

In the Introduction the inclusion of supporting references is needed. For instance, on lines 39 – 44 the authors fail to provide empirical support for statements regarding evidence of the role of food insecurity on student performance and quality of life. 

• RESPONSE: We have added the needed citations to the statements being made in this paragraph. 

On line 49 it is unclear as to what “rite of passage’ refers to. As Reviewer 1 queries “Is this referring to experiencing challenges accessing food as a “rite of passage.” This statement is vague and should be clarified. 

• RESPONSE: We have added “lack of access” to the statement about “safe, affordable, nutritious, and culturally relevant foods” to clarify what “rite of passage” is referring to. Experiencing a lack of access to food is considered by some to be a part of the university experience. Our research team encountered this belief when sharing the results from this study at our institution, with some arguing that their poor access to food was a part of their personal development as an undergraduate/graduate student. We decided to include the “rite of passage” statement to challenge the notion that food insecurity is a ‘normal’ part of the college experience. 

The authors should also check lines 55 – 56 (i.e., “There are is..”) and line 67 (i.e., Recent research explores…) for grammatical errors. 

• RESPONSE: Thank you for catching these errors. They have been corrected. 

More information is needed on the 10-item USDA Household Food Security Instrument. This is the instrument used to assess food security and more details are needed on how the measure is used and how it has been validated if not on a college student population as mentioned. This measure assesses the primary construct of interest and should be fully described. 

• RESPONSE: We have provided additional information about the 10-item USDA instrument, which includes references that discuss its evolution and use since its creation in 1995. We also cite the 2006 National Research Council (NRC) study, which is considered one of the most rigorous studies of the USDA instrument. Many of the recommendations in the NCR study were subsequently incorporated into the instrument by the USDA. The instrument is widely used in various national surveys, including the Current Population Survey (CPS) and is considered to be a valid and reliable instrument for measuring the food security of a population. 

In the Methods section on line 132 it states that “The word household was removed from the module” however, in Table 1 the word “household” is included. Please clarify. It is also unclear if Question 1 in the table is a screener question. It appears to be a screener question and if so, this should be stated. 

• RESPONSE: Thank you for catching this. We have revised Table 1 so that it now matches the survey text. We also added the word “Screener” to the first row of the table to indicate this text was the screener question. While the word household was removed from the screener question, it was retained, with qualifications, in the instrument. These qualifications were added to try and remove any uncertainty about what was meant by the term household. We have added additional text to the methods section to clarify the adjustments that were made.

A description of all measures assessed should be included in this section. For example, demographic questions pertaining to race/ethnicity and income are not included. There is also no mention of a measure or items that assess mental health. 

• RESPONSE: In the Methods sections we have added a description of the additional measures included in the survey and have cited the instruments/surveys that informed the mental health and housing questions. 

On line 167 the qualitative method used to analyze open-ended responses is provided. More information on this method and a reference is needed. What prompts or questions were provided to participants? Was inter-rater reliability assessed? Was qualitative software used to identify themes? On line 165 it is mentioned that “coding was conducted using Word”, this is vague. 

• RESPONSE: We have added additional detail on the qualitative analysis approach that we used. Following the method of Thomas and Harden (2008), we opted to engage in a group discussion of overlaps and divergences of qualitative codes rather than merely assessing the extent of consensus.

• Thomas, J., Harden, A. Methods for the thematic synthesis of qualitative research in systematic reviews. BMC Med Res Methodol 8, 45 (2008). https://doi.org/10.1186/1471-2288-8-45

The Results section could be strengthened if more details about the sample and measures are provided. Without more details (e.g., race/ethnicity) it is difficult to interpret the findings. What percentage of the sample is Black or African American? Has sample size been adjusted for small n’s? 

• RESPONSE: We have revised the S1 Table so that it now includes additional variables on race/ethnicity, gender identity, sexual orientation, and COVID-19. As mentioned in the methods section, a permutation chi square test was conducted when the sample size in any individual group was smaller than or equal to 5. 

The figures are blurred and difficult to see. Be sure to follow submission guidelines for submitting figures and graphs. 

• RESPONSE: We selected an appropriate R package to revise/recreate all of the figures in the manuscript. Since we needed to adjust how several variables were included in the analysis, a number of the findings have been revised. All of the figures have been processed using the PLOS ONE Preflight Analysis and Conversion Engine (PACE) digital diagnostic tool. 

The authors provide recommendations at the end of each result presented. It may be best to just present the findings in the Results section and in the Discussion section provide a summary of the findings with recommendations. 

In the Discussion section, qualitative responses are included. It is more appropriate to present these findings in the Results section as this is a mixed-methods study. As cited earlier, more details are needed on the qualitative component of the study and themes that emerged. 

• RESPONSE: We have revised the paper so that the qualitative results are now presented in the Results section and the recommendations are included in the Discussion section. 

Other comments relate to other areas the authors should consider. As the reviewer suggests, provide potential explanations to why the prevalence of food insecurity was lower in this study than in others. Knowing more about the university and campus locations students were recruited from may assist with this.

• RESPONSE: We have undertaken additional analysis on a survey question related to COVID-19 that we believe helps enrich our interpretation of the findings. Please see Fig 9, the related data in S1 Table, and the revised text below that speaks to the lower level of food insecurity found in this study. 

“The percentage of survey respondents found to have a low food security status (16%) is lower than what is typically found in studies at other comparable institutions of higher education (9,30,38). Following an initial survey of food access and security at the university (62), a food security taskforce was convened in 2020 that identified a series of barriers students face when accessing food and presented a range of recommendations and actions to address them (63). The most significant action taken was the creation of a free food service that could provide over 100 students a week with access to seasonal produce and other food items. This service began as a pilot program in 2020 and was formally launched in 2022 with additional donor support. In parallel with this, the university’s Dean of Students Office dedicated a portion of its emergency assistance fund to support students with food access needs via one-time grants of typically between $600-$800. 

Since 2019, the university has also significantly enhanced its communication of these support services, which also includes information about off-campus resources such as a local food pantry that is available to all students. In addition to the university’s actions, the Coronavirus Aid, Relief, and Economic Security Act (CARES Act) provided three rounds of direct relief payments to eligible adults and qualifying children, two of which were released during the year prior to the survey. 

While it is not possible to make a direct causal connection between the higher level of food security and the actions taken by the university or funding that may have been received via the CARES Act, written feedback from participants indicates that a number of students benefitted significantly from the university support they received. The higher level of food security could also be related to a self-selection bias due to COVID stressors. For example, students experiencing a decline in mental and physical health may not have had the capacity to complete the survey.”

How many of these students live on campus versus off campus? How many live with their family or a roommate as this too may aid in data interpretation. 

• RESPONSE: The S1 Table provides data on where students are living and how this corresponds with food security. 394 students lived on campus, 142 were living off-campus with their parents/guardians, 304 were living off-campus with their spouse/partner/dependents, and 1,271 were living off-campus by themselves or with roommates. Students living off campus with roommates were found to be 1.53 times more likely to have a low food security status when compared with students living on campus (see Fig 6).

Additionally, there seems to be vacillation between the term “low food security” and “food insecurity”. Are the authors suggesting a difference between the two terms? Are they the same? If so, it may be easier and more efficient to use one or the other. An operational definition of food insecurity might help. 

• RESPONSE: In an effort to minimize any confusion in our use of terms, we have removed reference to high/marginal and low/very low in the Results, Discussion and Recommendations, and Conclusion sections of manuscript, and now only use high and low food security. We have also made changes where possible to ensure food insecurity is used appropriately. 

Given that the study was conducted in April 2021 during the Covid-19 pandemic, this should also be considered as a potential factor influencing findings. In addition to participant recall, this too may be a potential confound and limitation.

• RESPONSE: Please see the previous response that discusses the new analysis we have undertaken on a COVID-19-related question. 

Overall, the topic is very interesting and has the potential to address factors associated with food insecurity on a college campus. As written, the paper is not accepted for publication but with significant revisions will be reconsidered.

• RESPONSE: We have completely revised the manuscript in response to the feedback provided. 

Reviewers' comments:

Reviewer's Responses to Questions

Reviewer #1: Introduction

Lines 39-44: Add references

• RESPONSE: We have added the needed citations to the statements being made in this paragraph. 

Lines 48-50: The wording of this sentence is a little unclear. It sounds like it’s saying that access to food is considered a “rite of passage.” Is this referring to experiencing challenges accessing food as a “rite of passage.”

• RESPONSE: We have edited this sentence to clarify the rite of passage statement. Please see the prior comment for additional information. 

Lines 55-56: This is the first time the 10-item USDA Household Food Security Survey instrument is mentioned. It would be helpful to briefly mention what this is and that it’s a commonly used measure for assessing food security status in the US.

• RESPONSE: We have added additional information on the 10-item USDA instrument, including that it is a module in the annual US Current Population Survey (CPS). Please see the prior comment for additional information. 

Lines 58-60: How does it allow for comparable results?

• RESPONSE: Since the majority of food security research at institutions of higher education are based on the USDA Household Food Security Survey instrument, using this instrument means that the research can be compared with these studies. We have revised the text to indicate that the 10-item USDA instrument has “been found to perform better than the 6-item version with regards to measuring food security among the college student population.”

Methods

Line 132: This says the word household was removed from the module, but the first question in table 1 uses household.

• RESPONSE: We have revised the text to indicate how the screener and subsequent questions of the 10-item USDA instrument were edited to ensure that students would be able to interpret the questions more clearly. Please see the prior comment for additional information. 

Since “high” and “low” are two of four USDA categories, I suggest referring to the two groups as food secure and food insecure as commonly done in other studies. The results section is confusing because it’s unclear if you’re referring to just the low food security group or low/very low combined. There is also inconsistency in the results section as to how it’s being referred to.

The methods section only describes the questions used to assess food security status. Briefly describe the other questions on the survey. There doesn’t need to be as much detail as for food security but some mention of other information being collected would be helpful.

• RESPONSE: As mentioned above, in an effort to minimize any confusion in our use of terms, we have removed reference to high/marginal and low/very low in the manuscript, and now only use high and low food security. We have also made changes where possible to ensure food insecurity is used appropriately. 

Lines 159-160: Was R used just for the odds ratios or also for the chi-square and Fisher’s exact tests described in the next paragraph. If it was used for all, you may want to move that sentence to the beginning or end of the information on statistical analyses to help clarify.

• RESPONSE: R was used for the odds ratios and the chi-square tests. The sentence was moved to the beginning of the paragraph.

Lines 167-169: How many coders were there and were responses double-coded or coded by a single coder?

• RESPONSE: Additional clarification has been provided in the Methods section. There were three coders who conducted simultaneous coding of the qualitative responses. A group discussion of overlaps and divergences was then conducted following the method of Thomas and Harden (2008).

Results

Lines 175 and 176: Be consistent in use of commas in numbers.

• RESPONSE: The manuscript has been edited to ensure we present numerical data in a consistent way.

Line 176: You may want to provide a little context about the university in the methods section as this is the first time having a main campus was mentioned. Many universities only have a single campus, so you may want to mention how many campuses there were in the methods and if the majority of students attended the main campus.

• RESPONSE: Table 3 provides descriptive data on sample and survey population. The majority (96.5%) of our students are based at the main campus in Blacksburg, and 93.5% of our respondents were based at this campus. We have added the following sentence to the methods sections that talks about our campuses: “The majority of the 2,116 respondents (94%) were enrolled at the university’s main campus in Blacksburg (Table 3). The university also has a presence in northern Virginia with several facilities located around the greater Washington, D.C., metro area, and has a small presence in Roanoke and Richmond, Virginia.”

Table 4: Rather than saying “by Selected Demographic Characteristics” you may want to say something like student classification.

• RESPONSE: We have revised the name of the table so that it better describes the data that is being presented. The new label for Table 3 is “Comparison of the Sample and Survey Population.” 

Line 220: The tables and figure seem to indicate that the question on being a first generation college student was a yes/no question and that data wasn’t collected to determine if a student was a second generation college student. Not all students who are not first generation college students are second generation – some may have grandparents, great grandparents, etc. who also attended college.

• RESPONSE: This is a great point and is something that we will endeavor to be more precise about in future research. The survey only asked respondents if they were a first-generation student. This question was connected with the First-Generation Student Support office at our university, which reaches out and communicates with students who meet the following definition: “At [... name of institution], a student is identified as a first-generation college student if neither parent/guardian has earned a bachelor’s degree at a four-year college or university.” Thus, respondents should know if they are a first-generation student based on this definition and the services they can access. Our analysis compares students who consider themselves to be first-generation students, with those who do not consider themselves to be first-generation students. 

Lines 235-236: I’m not sure that saying there is an important difference between the groups is the best way to reference that there was a large difference in the odds ratios. You may want to take out this sentence or reword to focus more on there being a difference in size of odds rations.

• RESPONSE: Agree. We have removed this statement. 

Discussion

Line 387: The way this is worded sounds like 7% of those who lived with their parents. Is this trying to say that 7% of students in the sample lived with their parents?

• RESPONSE: We have revised this text to make it clear that around 6% of the sample were living with their parents. 

Lines 475-476: You may want to briefly explain what work study is.

• RESPONSE: Additional text has been provided to explain how the Federal Work-Study (FWS) program functions.

Lines 480-515: There are many reasons why the prevalence of food insecurity was lower in this study than some others. You may want to mention some potential other reasons as well. This is a great example of a program the university is offering, but will a program reaching up to 115 students from a university of over 35,000 students likely be having a major impact on the prevalence of food insecurity on the campus?

• RESPONSE: Please see the previous response that discusses the new analysis we have undertaken on a COVID-19-related question.

Line 559: For future studies, you may want to keep in mind that there is both a 12-month and 30-day recall period for the 10-item US Household Food Security Survey Module.

Limitations

• RESPONSE: We have provided additional text in the introduction and methodology section about the 12-month and 30-day recall versions of the USDA 10-item Household Food Security Survey Module. This text indicates that the 30-day recall version is intended to study specific programs. We have added some additional text to the Limitations section which states that neither the 12-month or 30-day recall options may be ideal and that a revised instrument should be tested that does not refer to households and uses a recall period that aligns with the academic calendar. 

Another limitation to mention is external generalizability as this study included students from a single university

• RESPONSE: We have added the following sentence to the Limitations section: “Finally, since this study focuses on a single university, care should be taken when trying to generalize the findings to other institutions of higher education.” 

• RESPONSE: All of the figures have now been configured using PACE.

---

## [Editor Report · Decision Letter 1]

11 Dec 2023

PONE-D-23-05009R1Systematic investigation of inadequate food access at a large southeastern land grant universityPLOS ONE

Dear Dr. Hall,

Thank you for submitting your manuscript to PLOS ONE. After careful consideration, we feel that it has merit but does not fully meet PLOS ONE’s publication criteria as it currently stands. Therefore, we invite you to submit a revised version of the manuscript that addresses the points raised during the review process.

Once again, the authors are commended for addressing the topic of low food security among students in institutions of higher education. The revised manuscript addresses concerns raised in the previous review and is accepted with minor revisions. Most revisions are editorial. Recommended minor revisions are as follows: Introduction:  No recommendations provided. 

Methods: It is recommended that information about IRB compliance (i.e., The study was undertaken in compliance with the institutional research protocol IRB-21-122) be included after the description of the procedures (i.e., On or after Line 126).

Results: On line 221, use the term “Approximately” rather than “Around” and drop the word “both”.

• Line 227, delete “that were”.

• Lines 255 – 257. Was type of disability assessed? Did these students have a physical disability making it possibly more challenging to access food if appropriate resources were not available? This is an interesting finding as it confirms recent research on college students with disabilities and food insecurity (Stott GN, Taetzsch A, Morrell JS. College students with disabilities report higher rates of food insecurity. Disabil Health J. 2023 Oct;16(4):101485. doi: 10.1016/j.dhjo.2023.101485. Epub 2023 Jun 1. PMID: 37353371.). Understanding factors associated with student disability status and food insecurity could be an avenue for future research.

• Line 313, add the word “to” (i.e., mean students have access to the food they need due to dietary restrictions.)

• Line 356, remove the word “some” (i.e., but 13% of students were still…)

• Line 357, use the term “Approximately” rather than “Around”.

Discussion and Recommendations: On line 452 it is unclear if the recommendations provided address the preceding section in which it is discussed that first-generation or transfer students, GPA, disability status, and being Black/African-American predicts greater odds of low food security. The recommendations as stated are general and do not specifically address recommendations for these groups. While it is understood that strategies informing “first-year and existing students of the resources available to them” is important, providing specific recommendations or at best mentioning these groups who are potentially at-risk for low food security is important and reinforces the findings.

Other comments: Line 593 – Limitations and Direction for Future Research should  precede the section on Conclusions. The Conclusions should be listed last.

Again, the topic is very interesting and has the potential to address factors associated with food insecurity on a college campus. The authors have addressed previous concerns and have provided the recommended revisions. As mentioned, the paper should be accepted for publication with minor revisions.

We look forward to receiving your revised manuscript.

Kind regards,

Jocelyn Octavia Turner-Musa

Academic Editor

PLOS ONE
---

## [Author Response · Author response to Decision Letter 1]

21 Dec 2023

December 21, 2023

Response to Reviewers

Once again, the authors are commended for addressing the topic of low food security among students in institutions of higher education. The revised manuscript addresses concerns raised in the previous review and is accepted with minor revisions. Most revisions are editorial. Recommended minor revisions are as follows:

• RESPONSE: Thank you for reviewing the manuscript again. The updated manuscript addresses each of the comments below. 

Introduction: No recommendations provided. 

Methods: It is recommended that information about IRB compliance (i.e., The study was undertaken in compliance with the institutional research protocol IRB-21-122) be included after the description of the procedures (i.e., On or after Line 126). 

• RESPONSE: We have added the following sentence to the paper: “The study was undertaken in compliance with the institutional research protocol IRB-21-122.” Further, the research protocol, survey instrument, R code, and cleaned data set have been uploaded to the OSF site. This link will be made public if the paper is accepted: https://osf.io/bnuqv/?view_only=977d9eefa53d4e61b0ce61eada8cb048

Results: On line 221, use the term “Approximately” rather than “Around” and drop the word “both”.

• RESPONSE: We have made this revision. The sentence now reads: “Approximately 16% of undergraduate and graduate students can be classified as having low food security (Table 4).” 

• Line 227, delete “that were”.

• RESPONSE: We have made this revision. The sentence now reads: “The socio-demographic and economic characteristics linked to a higher likelihood of experiencing low food security included:”

• Lines 255 – 257. Was type of disability assessed? Did these students have a physical disability making it possibly more challenging to access food if appropriate resources were not available? This is an interesting finding as it confirms recent research on college students with disabilities and food insecurity (Stott GN, Taetzsch A, Morrell JS. College students with disabilities report higher rates of food insecurity. Disabil Health J. 2023 Oct;16(4):101485. doi: 10.1016/j.dhjo.2023.101485. Epub 2023 Jun 1. PMID: 37353371.). Understanding factors associated with student disability status and food insecurity could be an avenue for future research.

• RESPONSE: Thank you for sharing the above article and raising this point. Unfortunately, we did not capture the type of disability a student might have in order to maintain student privacy. If we run the survey again in 2025, we will consider collecting this information to build on the work of Stott et al. (2023). We have revised the text in this section as follows: “Students who self-assessed that they had a disability were 2.8 times more likely to have a low food security status when compared with students without a disability (Fig 1). The type of disability was not requested in order to maintain student privacy.” We have also added some new text on the need for research that understands the factors associated with student disability status and food insecurity to the Limitations section (see below).

• Line 313, add the word “to” (i.e., mean students have access to the food they need due to dietary restrictions.)

• RESPONSE: We have made this revision. The sentence now reads: “Further, having a dining plan does not mean students have access to the food they need due to dietary restrictions.”

• Line 356, remove the word “some” (i.e., but 13% of students were still…)

• RESPONSE: We have made this revision. The sentence now reads: “This group had one of the highest levels of food security (87.2%), but 13% of students were still experiencing food access problems (see S1 Table).”

• Line 357, use the term “Approximately” rather than “Around”.

• RESPONSE: We have made this revision. The sentence now reads: “Approximately 12% of the students living on campus had a low level of food security, compared with 17% of students living off campus by themselves or with roommates.”

Discussion and Recommendations: On line 452 it is unclear if the recommendations provided address the preceding section in which it is discussed that first-generation or transfer students, GPA, disability status, and being Black/African-American predicts greater odds of low food security. The recommendations as stated are general and do not specifically address recommendations for these groups. While it is understood that strategies informing “first-year and existing students of the resources available to them” is important, providing specific recommendations or at best mentioning these groups who are potentially at-risk for low food security is important and reinforces the findings. 

• RESPONSE: We have revised the text as follows to make sure the specific groups of impacted students are mentioned: “Developing strategies to inform first-year and existing students, especially those at higher risk of food insecurity (i.e., students with a disability, minority students, and low academically performing students), of the resources available to them throughout their studies is recommended. University administrations can work most effectively through student groups, associations, and initiatives that already reach students most at risk of experiencing food security to provide tailored information on available resources.”

Other comments: Line 593 – Limitations and Direction for Future Research should precede the section on Conclusions. The Conclusions should be listed last.

• RESPONSE: The Limitations section now proceeds the Conclusions. We have also added the following sentence that speaks to the need for more research on how students with disabilities could be better served: “While this research did not capture the type of disability a student might have in order to maintain student privacy, understanding the factors associated with student disability status and food insecurity is an important avenue for future research (64).”

Again, the topic is very interesting and has the potential to address factors associated with food insecurity on a college campus. The authors have addressed previous concerns and have provided the recommended revisions. As mentioned, the paper should be accepted for publication with minor revisions.

• RESPONSE: Thank you for taking the time to help us further improve the manuscript.

---

## [Editor Report · Decision Letter 2]

17 Jan 2024

Systematic investigation of inadequate food access at a large southeastern land grant university

PONE-D-23-05009R2

Dear Dr. Hall,

We’re pleased to inform you that your manuscript has been judged scientifically suitable for publication and will be formally accepted for publication once it meets all outstanding technical requirements.

Kind regards,

Jocelyn Octavia Turner-Musa

Academic Editor

PLOS ONE

Additional Editor Comments (optional):

Thanks for your resubmission. The article is accepted for publication and will contribute to empirical research on food insecurity on college campuses.
---

## [Editor Report · Acceptance letter]

13 Feb 2024

PONE-D-23-05009R2 

PLOS ONE

Dear Dr. Hall, 

I'm pleased to inform you that your manuscript has been deemed suitable for publication in PLOS ONE. Congratulations! Your manuscript is now being handed over to our production team.

Kind regards, 

on behalf of

Dr. Jocelyn Octavia Turner-Musa 

Academic Editor

PLOS ONE